# Biomarkers in Triple-Negative Breast Cancer: State-of-the-Art and Future Perspectives

**DOI:** 10.3390/ijms21134579

**Published:** 2020-06-27

**Authors:** Stefania Cocco, Michela Piezzo, Alessandra Calabrese, Daniela Cianniello, Roberta Caputo, Vincenzo Di Lauro, Giuseppina Fusco, Germira di Gioia, Marina Licenziato, Michelino de Laurentiis

**Affiliations:** Istituto Nazionale Tumori IRCCS “Fondazione G. Pascale”, Via Mariano Semmola, 53, 80131 Napoli NA, Italy; s.cocco@breastunit.org (S.C.); m.piezzo@breastunit.org (M.P.); a.calabrese@istitutotumori.na.it (A.C.); d.cianniello@breastunit.org (D.C.); r.caputo@breastunit.org (R.C.); dilaurovincenzo87@gmail.com (V.D.L.); g.fusco@breastunit.org (G.F.); germiradigioia@gmail.com (G.d.G.); m.licenziato@istitutotumori.na.it (M.L.)

**Keywords:** TNBC, *BRCA1/2*, HRR, PDL1, TILs, *PI3KCA*, *PTEN*, CTCs, CSC

## Abstract

Triple-negative breast cancer (TNBC) is a heterogeneous group of tumors characterized by aggressive behavior, high risk of distant recurrence, and poor survival. Chemotherapy is still the main therapeutic approach for this subgroup of patients, therefore, progress in the treatment of TNBC remains an important challenge. Data derived from molecular technologies have identified TNBCs with different gene expression and mutation profiles that may help developing targeted therapies. So far, however, only a few of these have shown to improve the prognosis and outcomes of TNBC patients. Robust predictive biomarkers to accelerate clinical progress are needed. Herein, we review prognostic and predictive biomarkers in TNBC, discuss the current evidence supporting their use, and look at the future of this research field.

## 1. Introduction

Triple negative breast cancer (TNBC) is a subtype of breast cancer lacking expression of estrogen receptor (ER), progesterone receptor (PgR) and human epidermal growth factor receptor 2 (HER2) [1]. TNBC accounts for approximately 10–15% of all breast cancers and is characterized by aggressive behavior, with trend to early relapse, metastatic spread, and poor survival [2]. Tumor heterogeneity of TNBC has been addressed as reason for different clinical outcomes and response to therapies. Lehmann et al. proposed a division of TNBCs into seven molecular subtypes: immunomodulatory (IM), mesenchymal (M), mesenchymal stem-like (MSL), luminal androgen receptor (LAR), unstable (UNS) subtype, and two basal-like subtypes (BL1 and BL2) [3]. Then, a subclassification refinement was performed, to define only four groups BL1 (immune-activated), BL2 (immune-suppressed), M (including most of the MSL), and LAR [4]. These classifications can, in theory, be used as prognostic and predictive tool for better patient selection and personalized treatments. For instance, LAR tumors, characterized by the expression of the androgen receptor (AR), are a subtype of TNBC that shares with Luminal (ER+) tumors some biological and clinical features [5,6,7] and are potentially sensitive to endocrine manipulation with AR antagonists.

However, sound clinical applications of this molecular classification are yet to appear [8].

In the last years, a great effort has been spent to identify new biomarkers and relative therapies, but only few of these have proven useful in clinical trials. Beside poly ADP-ribose polymerase (PARP) inhibitors, that have successfully been incorporated in the clinical practice in the BRCA1/2 subgroup of patients [9,10], and checkpoint inhibitor Atezolizumab, that was recently approved as front-line therapy in the metastatic setting [11], traditional chemotherapy without biomarker guide still remains the main therapeutic options for a large part of TNBC patients [12]. In this context, the implementation of more refined “omics” assays, along with appropriately designed clinical trials, may lead to the identification of new biomarkers to select new molecularly targeted therapies in TNBC. Several papers have provided a critical overview of numerous biomarkers evaluated in the past years in TNBC. In this review, we discuss the biomarkers that contributed to the development of new approved therapies in TNBC. We also review the new biomarkers that are showing promising results in ongoing clinical trials.

## 2. BRCA1/2 and Other Genes Involved in DNA Repair

The relationship between the tumor-suppressive genes BReast CAncer type 1 and Type 2 (*BRCA1/2*) and hereditary breast and ovarian cancer syndrome (HBOC) revolutionized clinical cancer genetics. First, the identification of a germline *BRCA1/2* mutations impacted cancer screening and prevention practices in this subgroup of patients and their relatives. Later, the knowledge of the pathological mechanisms of these mutations, led to the development of new therapeutic approaches, such as poly (ADP-ribose) polymerase (PARP) inhibitors, selectively directed on *BRCA 1* or *BRCA 2* deficient cells [13].

*BRCA1* and *BRCA2* are autosomal dominant and tumor suppressor genes involved in the preservation of genome integrity. Both genes play a crucial role in homologous recombination repair (HRR) of DNA. The *BRCA1* gene on chromosome 17q21 has a broader role than *BRCA2* in responding to DNA damage; it controls the signal transduction pathway involved in HHR, including recognition of genomic damage, checkpoint activation, recruitment of DNA repairing proteins, and decision of whether DNA double strand breaks (DSBs) needs to be resected; in addition, it is also involved in chromatin remodeling and transcription control [14,15]. The *BRCA2* gene on chromosome 13 plays the key role of recruiting the DNA recombinase RAD51 and localizing it to damaged DNA [16]. HRR is a conservative, error-free, mechanism of DNA repair due to its ability to restore the original DNA sequence. A small percentage of people (about one in 400, or 0.25% of the population) carry mutated *BRCA1* or *BRCA2* genes. However, compared to other subtypes of breast cancers, women with TNBC have a higher prevalence of germline BRCA mutations (gBRCAm), about 11–31% [17]. In addition to the well-known germline mutations, a smaller proportion of somatic mutations in *BRCA1/2* genes (sBRCAm) were also found in primary ovarian and breast carcinomas [18]. 

When *BRCA1/2* genes are defective, DNA damage is repaired by non-conservative mechanisms of DNA repair, such as non-homologous end joining (NHEJ), in order to maintain cell viability. This process of repairing DSBs is simpler than HHR and consists in joining the two broken DNA ends without the homologous DNA sequence to guide the repair: it is, therefore, prone to joining errors with mutation of the original sequence. In *BRCA1/2*-deficient cells, DNA DSBs repair is dependent on PARP1 protein [19,20]. PARP is an abundant, constitutively expressed nuclear enzyme that facilitates DNA repair, cellular proliferation, and signaling to other critical cell-cycle proteins and oncogenes. At sites of DNA damage, PARP activates intracellular signaling pathways that modulate DNA repair and cell survival [21]. Therefore, inhibition of PARP1 in *BRCA1/2*-deficient cells can lead to severe, highly selective toxicity in these cells [22]. This process has been called “synthetic lethality,” to highlight the interaction that occurs between the two genes when the perturbation of either gene alone is viable but the simultaneous perturbation of both genes results in the loss of viability [23].

In the past years, PARP inhibitors (PARPi) have been extensively studied as targeted therapy for gBRCAm ovarian and breast cancer patients. In 2014, the US Food and Drug Administration (FDA) approved the first PARPi, Olaparib, as monotherapy for patients with deleterious or suspected deleterious gBRCAm advanced ovarian cancer [24]. Later, other PARPi, such as nirapararib and rucaparib were approved [25,26,27,28]. On January 2018, based on data from OlympiAD trial (NCT02000622), FDA granted regular approval to Olpararib for the treatment of patients with deleterious or suspected deleterious gBRCAm, HER2-negative metastatic breast cancer previously treated with chemotherapy either in the neoadjuvant, adjuvant, or metastatic setting [9]. FDA also approved the BRACAnalysis CDx^®^ test (Myriad Genetic Laboratories, Inc., Salt Lake City, UT, USA), whose accuracy was established based on a retrospective/prospective analysis of the OlympiAD trial population. In the same year, FDA approved talazoparib for patients with deleterious or suspected deleterious gBRCAm, HER2 negative locally advanced or metastatic breast cancer, based on the phase III EMBRACA study results [10]. Based on promising results in the metastatic setting, PARPi are under investigation also in the early diseases. In particular, talazoparib is considered to be the most powerful PARPi candidate for single-agent treatment in neoadjuvant setting [29]. Promising results derived from a pilot trial of the MD Anderson Cancer Center (NCT03499353), in which gBRCA1/2m, HER2-negative, stage I-III BC patients, received neoadjuvant Talazoparib as single-agent during 4 to 6 months, without any chemotherapy, in order to evaluate the pathological complete response (pCR) rate and tolerance. The primary end point was residual cancer burden (RCB). Of 20 patients enrolled, 19 completed 6 months of treatment and 10 of them had pCR (RCB-0: 53%) while two additional patients had RCB-I. The rate of RCB 0-I was 63% overall, 57% in TNBC, and 80% in HR+, 53% in gBRCA1m, and 100% in gBRCA2m. Toxicities were managed by dose reduction and transfusions [30].

Other PARPis, such as Niraparib, Rucaparib, and Veliparib, are still under clinical evaluation in breast cancer both as monotherapy and in different combinations [31,32,33].

Several pre-clinical studies have shown that PARPis are able to inhibit cell growth and promote the death of breast cancer cells that are wild type for *BRCA1/2* [34]. A recent study assessed the efficacy of 13 different PARPis in the treatment of 12 different breast cancer cell lines that are either wild type or mutated for *BRCA1/2*. The TNBC cell lines MDA-MB-436 (*BRCA1*-deficient), MDA-MB231, and MDA-MB-468 resulted sensitive to Talazoparib, suggesting that the benefit of Talazoparib might extend to TNBC without *BRCA1/2* mutations [35]. This suggests that we should identify additional biomarkers for PARPis [36]. 

The term “BRCAness” has been used to describe a dysfunction in the BRCA-related DNA repair mechanism that is not due to mutations of the *BRCA1/2* genes. 

Deficiencies in a number of tumor-suppressor genes involved in HRR, such as ATM and ATR, may share the same therapeutic vulnerabilities with BRCAm tumors and confer sensitivity to PARP inhibition. Therefore, tumors with mutations in other HRR genes may also respond to a PARP inhibitor treatment [37].

On 2015, Domagala et al. performed genetic testing for 36 common germline mutations in genes engaged in HRR, (i.e., *BRCA1, BRCA2, CHEK2, NBN, ATM, PALB2, BARD1*, and *RAD51D*), in 202 patients, including TNBC and hereditary non-TNBC patients. As a result, 22.2% of 158 patients in TNBC group carried mutations in genes involved in DNA repair by HR [38]. Confirming data have been reported by other group [39] and it has been demonstrated that homologous recombination deficiency (HRD) can occur in sporadic cancer through genetic and epigenetic inactivation of other components such as *PALB2, BARD1, BRIP1, RAD51, RAD51C, RAD51D, ATM, FAAP20, CHECK2, FAM1, FANCE, FANCM, POLQ* [5,34]. These findings confirm the hypothesis that HRD TNBC, shares similar characteristics with gBRCAm TNBC, identifying new possible biomarkers of response to PARPi [40]. Instead of quantifying the effect of genetic variation in the HR pathway, researchers have developed methods to score the competency of the HR pathway. Three scoring systems have emerged: HRD-loss of heterozygosity (HRD-LOD), HRD-large-scale transition (HRD-LST), and HRD-telomeric allelic imbalance (HRD-TAI) [41,42,43].

Based on these findings, some clinical trials are now testing the use of PARPis in patients with BRCAm or HRD to maximize the number of individuals who may benefit from PARP inhibition [44,45,46]. The Phase II study Violette (NCT03330847) aims to assess the efficacy and safety of olaparib monotherapy versus two combinations (olaparib in combination with AZD6738 and olaparib in combination AZD1775), in TNBC patients prospectively stratified by presence/absence of qualifying tumor mutations in 15 genes involved in HRR pathway (*BRCA1, BRCA2, ATM, BARD1, BRIP1, CDK12, CHEK1, CHEK2, FANCL, PALB2, PPP2R2A, RAD51B, RAD51C, RAD51D*, or *RAD54L*). AZD6738 is an ATP competitive, orally bioavailable inhibitor of the Serine/Threonine protein kinase Ataxia Telangiectasia and Rad3 related (ATR), while AZD1775 is a small molecule WEE1 inhibitor used in combination with DNA-damaging agents in several trials [47]. Both, WEE1 and ATR, are kinases involved in the regulation of cell-cycle and DNA repair. Preclinical in vitro and in vivo experiments demonstrate that AZD1775 has synergistic cytotoxic effects when administered in combination with PARPis, while cancer cells with defective DNA repair mechanisms or cell cycle checkpoints may be particularly sensitive to ATR inhibition; this has led to the development of early-phase trials on three ATR inhibitors (M6620, AZD6738, and BAY1895344) [48,49].

Somatic BRCA mutations (sBRCAm) were supposed to report a phenotype similar to tumors from patients with germline mutations and response to PARPi [50]. A recent metanalysis, including 236 patients, with sBRCAm and 1204 patients with gBRCAm treated with PARPi for different cancers, indicates similar response rates of PARPi therapy [51]. In ovarian cancer, where the rate of somatic mutation is higher compared to the other cancers, PARPi have shown efficacy in patients carrying mutations of *BRCA1/2*, either germline or somatic, but also in wild-type *BRCA1/2* [52]. In breast cancer, clinical trials evaluating the predictive role to PARPi of sBRCAm are ongoing (NCT03990896, NCT03286842, NCT04053322, NCT03344965, NCT03920488, NCT01434420, NCT03078036). To the last ASCO meeting were presented the results from TBCRC 048 trial (NCT03344965), a phase II study of olaparib monotherapy in 54 metastatic breast cancer patients, of which 40 patients ER+ HER2−, 3 HER2+, and 10 TNBC, divided in 2 cohorts based on germline mutations in non-*BRCA1/2* DDR-pathway genes (cohort 1) and on somatic mutations in these genes or *BRCA1/2* (cohort 2). Results showed an ORR of 29.6% in Cohort 1 and 38.5% in Cohort 2, with *sBRCA1/2* or *gPALB2* mutations predictor of response in this last one [53].

Since *BRCA1/2*-deficient tumors have defects of DNA repair, the use of agents causing DNA damage have been extensively investigated as candidate therapy to promote mechanisms of cell cycle arrest and apoptosis in these tumors. Platinum drugs, causes platination of genomic and mitochondrial DNA, forming intrastrand crosslink adducts which cause double-strand breaks that culminate in the activation of apoptosis, especially when DNA lesions cannot be repaired [54]. This suggests that BRCA mutational status is a promising biomarkers for platinum-based chemotherapy. In metastatic setting, the 4th ESMO guidelines recommend anthracycline-taxane chemotherapy as first line for treatment of advanced TNBC, while carboplatin may be considered for BRCA positive TNBC as second line treatment [55]. Regarding sensibility to platinum agents of BRCAness phenotype in this setting, the phase II TBCR009 trial has shown a correlation between high HRD scores and their predictive response to platinum-based chemotherapy, beyond *BRCA1/2* mutations, in advanced first and second line TNBC patients. In particular, ORR was 54.5% in patients with germline *BRCA1/2* mutations, while, in patients without *BRCA1/2* mutations, HRD-LOH/HRD-LST scores discriminated responding and nonresponding tumors (12.68 and 5.11, respectively). Five of the six long-term responders alive at a median of 4.5 years lacked germline *BRCA1/2* mutations, and two of them had increased tumor HRD-LOH/HRD-LST scores [56]. At the last ASCO 2020 meeting, were presented results of SWOGS1416, a phase II randomized trial of cisplatin +/− veliparib in metastatic TNBC and/or germline BRCA-associated breast cancer (NCT02595905). In this study 248 TNBC patients were classified into the three groups (1) 37 gBRCA+ (2) 101 BRCA-like (3) 110 non-BRCA-like, based on results of central gBRCA testing and of multi-pronged biomarker panel including myChoice HRD score, somatic *BRCA1/2* mutations, *BRCA1* methylation, and non-*BRCA1/2* HR germline mutations. Results showed that addition of Veliparib to cisplatin significantly improved PFS and numerically improved OS and ORR, of BRCA-like sub-group of patients, that included 76% of HRD ≥ 42, 17% of *BRCA1* promoter methylation, and 7% of HRR genes mutations [57].

In early settings, several neo-adjuvant clinical trials have evaluated the impact of adding platinum to standard chemotherapy, however, its use remains controversial and it is not routinely recommended in unselected TNBC or BRCA mutations carriers [58]. The neoadjuvant phase 2 trial of cisplatin in TNBC by Silver et al. showed a pCR rate of 22% among all TNBC patients (n = 28). Between these, germline *BRCA1*, low *BRCA1* mRNA expression or *BRCA1* promoter methylation patients achieved pCR [59]. The GeparSixto trial, which included stage II or III TNBC and HER2+ patients, demonstrated significant improvement in pCR with carboplatin added to an anthracycline/taxane-based neoadjuvant chemotherapy in TNBC patients [60]. This study tried also to prove a possible correlation between BRCAness phenotype and response to platinum. The exploratory endpoints of the trial investigated (1) whether the HRD assay predicts specifically for carboplatin response or independently of treatment and (2) whether there is an association of HRD with long-term outcome in patients with BRCA mutations detected in tumor tissue (tmBRCA) and in patients with high HRD without tmBRCA. Results showed that the HR deficiency (defined as HRD score 42 and/or presence of tmBRCA) and HRD score in non-tmBRCA were predictors of response, however, the HRD assay failed to identify a subset of patients most likely to derive benefit from the addition of carboplatin [61]. In this context, the PrECOG 0105 phase II study of gemcitabine and carboplatin plus iniparib as neoadjuvant therapy for TNBC and *BRCA1/2* mutation-associated breast cancer, revealed that combination of gemcitabine, carboplatin, and iniparib is active in the treatment of early-stage triple-negative and *BRCA1/2* mutation-associated breast cancer. In particular, responder patients with sporadic TNBC, lacking *BRCA1/2* mutation, had an elevated HRD-LOH score, suggesting that HRD assay could predict to platinum response [62]. In a recent metanalysis including seven studies, with a total of 808 TNBC patients, among which 159 BRCA mutated, was reported that the addition of platinum to chemotherapy regimens in the neoadjuvant setting increases pCR rate in BRCA mutated as compared to wild-type TNBC patients, however, this trend did not achieve statistical significance [63]. Recently, through a whole genomic sequencing technique, researches developed a new weighted model called HRDdetect with a sensibility of 98.7% to detect *BRCA1/2* deficient samples [64]. Elevated HRDetect was significantly associated with clinical improvement on platinum-based therapy in advanced breast cancer [65].

Preclinical data support a potential synergism between PARPi and platinum-based chemotherapy [66,67]. The addition of low-dose veliparib (VELI) to carboplatin–paclitaxel versus placebo was tested in HER2-negative, gBRCA1/2m, advanced BC (BROCADE 3 trial). Veliparib increased PFS while no impact on OS was reported [68]. In the neoadjuvant setting, Veliparib was assessed in the carboplatin-VELI arm of the randomized phase II ISP-Y 2 trial. In this trial, the carboplatin-VELI arm enrolled 72 patients with HER2-BC, who received Veliparib plus carboplatin during the paclitaxel sequence (VELI-CARBO) followed by doxorubicin and cyclophosphamide (AC). The benefit of VELI-CARBO seemed to be restricted to the TNBC patients (51% pCR for VELI-CARBO vs. 26% pCR in the control arm in TNBC) [69]. The randomized phase III trial (BrighTNess) was subsequently conducted in 634 TNBC patients receiving neoadjuvant chemotherapy evaluating VELI-CARBO vs. placebo-CARBO or double placebo, in combination with paclitaxel followed by four cycles of AC. The pCR rate was higher in the CARBO-containing regimen (53% in VELI-CARBO and 58% in the placebo-CARBO vs. 31% in the control arm). Within the patients with gBRCA1/2m, no significant differences were achieved, the pCR rate was 57% in the VELI-CARBO/paclitaxel arm, 50% in the placebo-CARBO/paclitaxel arm, and 41% in the control paclitaxel arm. In summary, the trial did not detect statistically significant differences between these subgroups [70].

Research is also focusing on biomarkers of resistance to PARPis. Acquired or innate resistance to single-agent PARPis has been frequently observed in both preclinical and clinical studies [71,72,73,74]. The main mechanisms described are the reversion of BRCA and HRR gene mutations to wild-type, the demethylation of promoter of HR genes, the mitigation of replication stress, the mutations in PARP itself and/or drug efflux pumps [75,76,77]. In preclinical patient-derived BRCAm-xenograft (PDX) models, the detection of RAD51 foci, a surrogate biomarker of HRR functionality, correlated with resistance to PARPis regardless of the underlying mechanism restoring HRR function [78,79,80,81,82]. By using in vitro and in vivo models of intrinsic resistance to PARPis, Yu-Yi Chu et al. described the role of proteins like RTK, c-MET in PARP inhibition resistance [83,84].

The identification of biomarkers of resistance can drive the research of pharmacological strategies to delay or prevent the development of the drug resistant phenotype. Emerging data suggest that olaparib-resistant cancer models can be re-sensitized to olaparib when combined with AZD1775 or AZD6738 [85,86,87], leading to early phase clinical trials combining ATR inhibitors and PARPis in different cancers (NCT02723864, NCT03462342, NCT03682289, NCT02576444).

## 3. Biomarkers of Immunotherapy in TNBC

It is well-known that the immune system and cancer have a complex interplay: it is a multi-step process, named cancer immunoediting, largely mediated by CD8+ cytotoxic T lymphocytes, and in which both immune-stimulatory and inhibitory factors are involved. During the first phase, called elimination phase, the innate and adaptive immune system recognize and reject tumor cells, then the surviving tumor subclones can progress into a state of dormancy, the equilibrium phase, in which tumor growth is limited and tumor cells are gradually selected through upregulation of pro-survival pathways, changes in expression of molecules involved in immune suppression or angiogenesis. These immune-edited tumor cells can then enter into the escape phase, in which the tumor growth is uncontrolled [88].

Breast tumors have been historically considered non-immunogenic diseases, with a relatively low mutation rate. However, among BC subtypes, TNBC is characterized by high mutation rate and greater tumor-infiltrating lymphocytes (TILs). TILs are present both intratumorally and in adjacent stromal tissues and are composed mainly of cytotoxic CD8+ lymphocytes, and, to a lesser extent, CD4+ T-helper cells, T-regulatory (Treg) cells, macrophages, mast cells, and plasma-cells. 

The presence of intra-tumoral and stromal TILs has predictive and prognostic role; in TNBC increased TILs at diagnosis have been associated with pathologic complete responses with neoadjuvant chemotherapy and improved survival after adjuvant chemotherapy [89,90,91]. The association between high number of stromal TILs and more favorable survival outcomes, in terms of overall survival (OS) and disease-free survival (DFS) highlighted the prognostic value of immune antitumoral activity, while the association between high TILs and response to chemotherapy established the predictive value of TILs as marker of response to chemotherapy. This also suggests that the effect of chemotherapy may be partially mediated by the immune system, making the investigation of immunotherapy in TNBC particularly interesting [92,93,94]. 

Programmed cell death-1 (PD-1) is an immune checkpoint able to inhibit both adaptive and innate immune response, and is expressed on the surface of immune cells, such as T cells, B cells, natural killer (NK), macrophages, dendritic cells (DCs), and monocytes [95]. PD-1 controls induction of tolerance to antigens and termination of immune response, playing a key role, under physiological condition, in maintaining the immune tolerance and limiting the autoimmunity. In the tumor microenvironment PD-1 is involved in the development of tumor immuno-tolerance [96,97]. PD-1 ligand, programmed cell death-ligand 1 (PD-L1), is a transmembrane protein expressed both on tumor cells and immune cells (DCs, B cells, T cells, macrophages), and it represents an “adaptive immune mechanism” that cancer cells may use to escape anti-tumor immunity. 

PD-1/PD-L1 ligation acts as pro-tumorigenic pathway, leading to deactivation of T-cell function and resulting in the escape from immune surveillance [98,99,100,101,102]. PD-1/PD-L1 expression can be regulated by various signals in cancer cells, such as (1) activation of PI3K/AKT pathway that promote the expression of PD-L1 through increased extrinsic signaling and by downregulation of PTEN [103]; (2) MAPK signaling pathway, involved in the conversion of extracellular signals into intracellular responses and associated to PD-1/PD-L1 axis [104,105]; (3) JAK-STAT signaling pathway that provides a key mechanism for extracellular signals to control gene expression, including the expression of PD-L1 [106]; (4) abnormal WNT signaling pathway, able to interfere with cancer immuno-monitoring and to promote immune escape by a crosstalk mechanism between WNT activity and PD-L1 expression [107,108]; (5) NF-κB signaling pathway that mediates INF-γ-induced PD-L1 expression [109,110]; (6) Hedgehog signaling pathway that promotes the expression of PD-1/PD-L1 axis and which inhibition may induce lymphocyte antitumor activity [111]. 

PD-L1 is commonly expressed in 20% of TNBC and has been related to distinctive characteristics of BC, such as younger age, large tumor size, high grade, high proliferation, ER-negative status, and HER2-positive status. PD-L1 is expressed on about 10% of tumor cells (TC), while its expression on tumor infiltrating immune cells (IC) is higher (40–65%). The predictive role of PD-L1 positivity on IC has been validated in several clinical trials. The expression of both PD-1 and PD-L1 is associated with a good outcome and it is correlated with better overall survival and higher sensitivity to chemotherapy, confirming that the cytotoxic effect of chemotherapy is partially mediated by the immune response against tumor [112,113,114,115,116]. The expression of PD-L1 on tumor infiltrating immune cells (IC) may also play a role as biomarker as it has been shown in several clinical trials [117,118,119].

The introduction of immune checkpoint inhibitors (PD-1 inhibitors and PD-L1 inhibitors) as strategy to wake up the immune cells and reduce the tumor growth, is playing a critical role in improving treatment of TNBC. Atezolizumab (anti-PD-L1) is the first in class receiving the FDA accelerated approval on March 2019, based on results from IMpassion 130 study that showed a significant OS improvement (25 months vs. 18 months) in patients with IC PD-L1 expression (PD-L1+ IC ≥ 1%), treated with Atezolizumab plus nab-paclitaxel versus nab-placlitaxel alone as first line therapy for metastatic TNBC [11,120]. FDA also approved the VENTANA PD-L1 (SP142) assay as a companion assay to determine PD-L1 expression on IC. The analytical power of the VENTANA assay was evaluated and compared with other two immunohistochemistry assays (22C3 and SP263) in a post-hoc exploratory analysis of IMpassion 130 study. Overall, the VENTANA assay was predictive of atezolizumab efficacy when performed either on the primary or on the metastatic tumor specimen. With the cutoff of ≥1% of PDL+ IC, it identified a smaller population of pts as compared with 22C3 and SP263 assays, but with a higher predictive performance [121]. 

In the same setting, results from Keynote-355 trial, evaluating the combination of Pembrolizumab (anti-PD-1) plus chemotherapy as first line treatment of metastatic TNBC, have lately been presented at ASCO meeting. This study confirms the importance of evaluating PDL1 expression as biomarker. PDL1 has here been evaluated with the 22C3 (DAKO PharmaDx) assay using the combined positive score (CPS) defined as which is the number of PD-L1 staining cells (tumor cells, lymphocytes, macrophages) divided by the total number of viable tumor cells, multiplied by 100. Indeed, pembrolizumab has shown to improve progression-free survival (PFS) only in patients with a CPS score ≥10 [122]. Of note, however, that the predictive value of the same assay had not previously confirmed in the Keynote-086 study, with Pembrolizumab used as monotherapy: in the cohort A of the study (enrolling previously treated metastatic TNBC patients), PFS was similar irrespective of PD-L1 expression status [123].

Finally, in neo-adjuvant setting, conflicting results from Keynote-522 and NeoTRIPaPDL1 studies, investigating the combination of standard chemotherapy with Pembrolizumab and Atezolizumab respectively, showed that the role of PD-L1 expression as predictive marker of response to immune checkpoint inhibitors is still controversial [124,125]. Particularly, matters of uncertainty are: What is the best assay; what is the best cutoff to define PDL1-positivity; what drugs (pembrolizumab vs. atezolizumab vs. others) these apply to; what disease setting (early vs. metastatic) these apply to. 

Evaluating other predictive biomarkers, in addition to PD-L1 testing, may help to select patients who could benefit from ICI. 

Microsatellite instability (MSI) is a hypermutable phenotype generally deriving from a deficit in the DNA mismatch repair mechanism (deficient mismatch repair; dMMR). MSI is evaluated by identifying mutations involving microsatellites located throughout much of the genome as short DNA sequences repeated in tandem. High levels of microsatellite instability (MSI-H), corresponding to dMMR, were found across several cancers, such as endometrial and gastrointestinal cancers (20%–30%). MSI-H is correlated to a high neoantigen burden and, therefore, to a high immunogenic potential and sensitivity to immuno-checkpoint inhibitors, irrespective of the tumor histologic type. MSI-H/dMMR was the first biomarker to grant a “site-agnostic” FDA approval to an anticancer drug: on October 2016, based on the results of the Keynote 158 trial, Pembrolizumab was approved for use as monotherapy on solid tumors with MSI-H or dMMR and without satisfactory therapeutic alternatives [126]. While the frequency of MSI-H/dMMR in BC is very low (0%–1.5%) and its use as prognostic or predictive biomarker is still under investigation [127,128,129], the FDA approval allows using Pembrolizumab for TNBC with MSI-H/dMMR.

Tumor mutational burden (TMB), calculated as the total number of mutations in a sample divided by the length of the genomic target region (mut/Mb), is a good marker of tumor antigenicity. Since the greater number of somatic mutations, it is probable that these mutations will yield to misfolded proteins (neoantigens) capable of being immunogenic and providing targets for T-cell response. In several tumors, such as lung cancer, melanoma, and colorectal cancers, TMB, easily evaluated by NGS techniques, represents a good predictive biomarker for ICI response, since high TMB is associated with high neoantigen burden, high T-cell infiltration and high response rate to ICI, regardless the PD-L1 status [130,131]. In BC, the predictive role of TMB is still controversial, recent data showed that overall 3.1%–5% of breast cancers are hypermutated, with high prevalence in TNBC and metastatic tumors; these tumors seem to be more likely sensitive to PD-1 inhibitors after a preliminary analysis of clinical and genomic data, also if no differences in terms of survival has been shown in patients with high TMB treated with ICI [132,133,134]. In neoadjuvant setting, the GeparNuevo trial, investigating the addition of Durvalumab (anti-PD-L1) to anthracycline/taxane-based chemotherapy, showed a significant trend for increased pCR rates for PD-L1-positive patients in both PD-L1-TC in Durvalumab arm and PD-L1-IC in placebo arm. In addition, they performed a predefined analysis of 149 samples assessing the predictive value of TMB alone or in combination with an immune gene expression profile (GEP) for pCR. Results from multivariate analysis showed an odds ratio for pCR per mut/Mb of 2.06 (95% CI 1.33–3.20, P = 0.001) among all patients, 1.77 (95% CI 1.00–3.13, P = 0.049) in the Durvalumab treatment arm, and 2.82 (95% CI 1.21–6.54, P = 0.016) in the placebo treatment arm, confirming that further analyses of TMB in combination with other immune parameters are still necessary, as well as the choice of a standardized assay and a cut off value to define high mutational load [135,136]. Very recently, FDA approved FoundationOne^®^ CDx test to identify patients with solid tumors with TMB score who may benefit from immunotherapy treatment with Pembrolizumab monotherapy. FoundationOne CDx is a next-generation sequencing-based in vitro diagnostic device for detection of substitutions, insertion and deletion alterations (indels), and copy number alterations (CNAs) in 324 genes and select gene rearrangements, as well as genomic signatures including microsatellite instability (MSI) and tumor mutational burden (TMB) using DNA isolated from formalin-fixed paraffin-embedded (FFPE) tumor tissue specimens. The accelerated approval was based on data from a prospectively planned retrospective analysis of 10 cohorts of patients with various previously treated unresectable or metastatic solid tumors, who were enrolled in KEYNOTE-158 (NCT02628067), a multicenter, non-randomized, open-label trial evaluating Pembrolizumab (200 mg every three weeks). TMB status was assessed using the FoundationOne CDx assay and TMB-High (TMB-H) was defined as TMB ≥10 mut/Mb. The results showed that patients with TMB-H in solid tumors who were treated with Pembrolizumab had a higher overall response rate (29%) compared to patients with TMB [137].

Despite the promising results of PD-1/PD-L1 axis blockade, additional strategies to improve the response rate and to overcome resistance to immunotherapy are ongoing, such as testing novel immunomodulatory compounds, which might increase the activity of immunotherapy treatment and contribute to convert “cold tumors” into “hot tumors.” This novel combinations are focusing on the dual immune checkpoint blockade, which includes the introduction of antibodies against co-inhibitory, such as anti-LAG-3 antibodies.

Lymphocyte-associated gene 3 (LAG3) is a transmembrane protein with structural homology to the CD4 co-receptor and mainly expressed in activated CD4+ T cells, T-regulator cell, Tr1 cells, activated CD8+ T cells, natural killer cells, dendritic cells, B cells, and exhausted effector T cells. LAG3 negatively regulates the proliferation, activation, and effector function of T cells [138]. The role of LAG3 as prognostic biomarker is still controversial and under investigation. Results from a recent meta-analysis investigating the role of LAG3 as prognostic biomarker in several solid tumors, including TNBC, showed that high expression of LAG3 can be associated with favorable outcome, particularly in early stage tumor, also if there was a borderline statistical significance in their results, suggesting that the role of LAG3 as prognostic biomarker should be evaluated together with the expression of other biomarkers reflecting an active host immunity, such as PD-L1 and CD8 [139].

The suggestion that other molecules can stimulate and increase the number of immune cells, enhancing the anti-tumor immune function, is leading to new combination approaches of immune checkpoint inhibitors and novel agents for the treatment of TNBC. The InCITe trial (NCT03971409) is evaluating the combination therapy of Avelumab (anti-PD-L1) and Binimetinib (MEK 1/2 inhibitor), Utomilumab (anti-4-1BB), or PF-04518600 (anti-OX40) in a multi-arm study for treatment of first or second line in metastatic TNBC patients.

## 4. PI3KCA and PTEN Mutations as Predictive Biomarkers

The phosphatidylinositol 3-kinase (PI3K) pathway is a key regulator of survival, growth, proliferation, angiogenesis, metabolism, and migration. It comprises a family of intracellular signal transducer enzymes with three key regulatory nodes, PI3K, AKT, and mammalian target of rapamycin (mTOR). PI3K activation phosphorylates and activates AKT, that regulates the functions of numerous cellular proteins, including the FoxO proteins, mTOR complex 1 (mTORC1), and S6 kinase [140,141]. In many cancers, this pathway is overactive due to *gain*-*of-function* mutations of phosphatidylinositol-4, 5-bisphosphate 3-kinase, catalytic subunit, alpha (*PIK3CA*), *loss-of-function* alterations of the tumor suppressor phosphatase and tensin homolog (*PTEN*), deregulation of receptor tyrosine kinase signaling, and amplification and mutations of receptor tyrosine kinases [142,143]. These alterations occur in approximately 35% of triple-negative and 45% of ER/PgR-positive, HER2-negative breast cancers [144]. PI3Ks are a family of lipid kinases that are divided into three classes based on their structures and substrate specificities. Class IA PI3Ks are heterodimers that contain a p110 catalytic subunit and a p85 regulatory subunit. The genes *PIK3CA*, *PIK3CB*, and *PIK3CD* encode three highly homologous class IA catalytic isoforms: p110α, p110β, and p110δ, respectively [145]. In TNBC, the majority of activating mutations occur in the p110a (alpha subunit encoded by *PIK3CA*), overall mutated in ~9% of primary TNBC. *PIK3CA* mutations result in activated alpha PI3K, leading to an activating downstream pathway. *PTEN* alterations are also frequent in TNBC, with genetic loss of function occurring in 15% [146,147]. The phosphatase PTEN exert its activity of tumor suppressor through dephosphorylating phosphatidylinositol (3,4,5)-trisphosphate (PIP3), resulting in an inhibition of AKT [148,149]. Inactivating mutations of *PTEN*, including truncating and frameshift mutations or homozygous deletion, cause loss of function with consequent hyperactivation of AKT activity. Single nucleotide variants hotspots mutations such as R130X, R233X, and R335X, allelic loss in loci of the10q23 region were also reported [147,150,151]. A single amino acid substitution E17K of AKT1 was described in several cancers, with the highest incidence 3.8%, in breast cancer. This mutation results in a pathologic association of AKT1 with the plasma membrane and its constitutive activation [152]. Finally, mutations in *mTOR* were found in 1.8% of breast cancer [153]. The high frequency of mutations of PI3K/AKT/mTOR pathway found in breast cancer provides the rationale to test new inhibitors in combination with standard therapies. Several PI3K and AKT inhibitors are currently under investigation in clinical trials, mostly in ER/PgR-positive HER2-negative subtypes. Despite intense research efforts, so far, only *PIK3CA* mutations have proven to have a predictive value for treatment with α-selective and β-sparing PI3K inhibitors, Alpelisib and Taselisib respectively, in the advanced setting [154,155,156,157,158]. Alpelisib, an oral α-specific PI3K inhibitor that selectively inhibits p110α, was recently approved, based on results of phase III SOLAR-1 trial (NCT02437318), for postmenopausal women, and men, with metastatic or advanced *PIK3CA*-altered, ER/PgR-positive, and HER2-negative breast cancer, indicating that the integration of genomic testing for *PIK3CA* mutation may be useful in the selection of therapy [154,155].

The predictive effect of *PIK3CA* mutations may also have future relevancy for TNBC. Recently, Yuan et al. showed that combination of CDK4/6 inhibitor Ribociclib and Alpelisib caused the reduction of p-RB and p-S6, of MCL-1, induction of apoptosis, and an enhanced reduction of tumor growth in a TNBC PDX model [159]. Other findings reported that same combination significantly increased tumor-infiltrating T-cell activation and cytotoxicity and decreased the frequency of immunosuppressive myeloid-derived suppressor cells in a syngeneic TNBC mouse model [160]. These studies support the development of new possible combinational approaches in TNBC. 

LAR TNBC, in particular, are sensitive to endocrine manipulations with AR antagonists and are enriched (approximately 40%–50%) in activating *PIK3CA* mutations [161,162]: this could confer sensitivity to PI3K inhibitors and synergy with AR antagonists. Based on these findings, in the TBCRC 032 IB/II trial (NCT02457910) an oral antiandrogen, enzalutamide, has been evaluated with or without taselisib in patients with AR+ metastatic TNBC. By RNA seq analysis, the authors noticed that patients receiving the combination displayed decreased expression of genes involved in mTOR signaling and increased expression of genes related to adaptive immunity after treatment. Overall this study demonstrated that this combination can be given safely and appears to increase clinical benefit in TNBC patients with AR+ tumors [163]. 

Loss of PTEN tumor suppressor activity has been investigated as biomarker of response to AKT inhibitors, based on finding that *PTEN*-loss could enhance activation of AKT signaling. The randomized Phase II study (GO29227, LOTUS NCT02162719) compared activity of Ipatasertib [164,165,166,167], a potent, highly selective inhibitor of all three isoforms of Akt, plus Paclitaxel versus placebo plus Paclitaxel as first-line treatment for patients with inoperable locally advanced or metastatic TNBC. Patients were classified according to PTEN expression by immunohistochemistry (PTEN-low or PTEN-high) and *PIK3CA/AKT1/PTEN* genomic alterations (*PIK3CA*, *AKT1* or *PTEN* mutations) characterized by NGS. Results have shown that the increase in median PFS was quite modest in the Intention-to-treat population and PTEN-low subgroup but more pronounced in predefined analyses of the patient population with *PIK3CA/AKT1/PTEN*-altered tumors, suggesting that a complete assessment of PI3K pathway could have a predictive value rather than a single alteration [168]. 

At present, Phase III Ipatunity 130 trial (NCT03337724) is aiming to confirm data from LOTUS trial, evaluating Ipatasertib in combination with Paclitaxel in *PIK3CA/AKT1/PTEN*-Altered, locally advanced or metastatic, TNBC or ER/PgR-positive, HER2-negative patients. Genetic alterations will be evaluated in relevant genes in the PI3K/Akt pathway, both in tissues and in blood samples, by NGS assay. 

In the neoadjuvant setting, the phase II FAIRLANE study (NCT02301988), evaluating Paclitaxel plus Ipatasertib or placebo in TNBC, partially supports the potential utility as biomarker for the *PIK3CA/AKT1/PTEN* alterations; there was, indeed, in the trial, a numerically but non-significant increase in pCR rates, with more pronounced results in patients with PTEN-low tumors (32% versus 6%) and *PIK3CA/AKT1/PTEN*-altered tumors (39% versus 9%) [169]. In contrast, in the phase Ib study NCT03800836, evaluating the efficacy and safety of the combination of Ipatasertib, Tecentriq (Atezolizumab), and chemotherapy (Paclitaxel or Nab-paclitaxel) as a first-line treatment option for people with advanced TNBC, the objective response rate (ORR) was 73% (95% CI 53–88%), irrespective of tumor biomarker status [170,171]. A confirmatory phase III study NCT04177108 investigating the combination of ipatasertib, atezolizumab and paclitaxel as first-line therapy for locally advanced/metastatic TNBC cancer is still ongoing. 

The predictive potential of *PIK3CA/AKT1/PTEN* alteration may is also supported by other AKT inhibitors trials. In the PAKT trial, Capivasertib (AZD5363), a potent and selective oral inhibitor of all three isoforms of the serine/threonine kinase AKT [172] has been evaluated in TNBC, showing that addition of capivasertib to first-line paclitaxel therapy prolonged PFS and OS. Also in this case, benefits were more pronounced in patients with *PIK3CA/AKT1/PTEN*-altered tumors [173].

In summary, alteration of PIK3CA/AKT1/PTEN pathway have not yet satisfied criteria for the clinical use in TNBC, but diverse evidences support further research on this topic.

## 5. Promising Molecular Biomarkers

### 5.1. New Targets of Antibody–Drug Conjugates in Triple Negative Breast Cancer

Antibody drug conjugates (ADCs) are a new class of anticancer drugs that share the same general mechanism of action; they are designed as a monoclonal antibody that are conjugated with a potent cytotoxin (so called payload). The monoclonal antibody is directed against an antigen on the surface of the target cancer cell, and upon binding to the target antigen, they are internalized and release the payload inside the cell, leading to selective cytotoxicity. This allows selective intracellular delivery of very potent payload that, because of their inherent toxicity, could not be infused as free molecules to the patient. As an additional mechanism of action, ADCs can elicit a potent immune response by inducing dendritic cell maturation and CD8 and CD4+ T-cell infiltration [174,175].

For an ADCs to be effective, a critical factor is the target antigen, that has to be selectively expressed (or, overexpressed) on the intended cancer cell. Therefore, the presence (or the overexpression) of the target antigen can be tested as biomarker to identify potentially sensitive patients. Several molecules have been identified in TNBC cells that meet these characteristics. The most promising ones are: (1) the glycoprotein non-metastatic b (GPNMB); (2) trophoblast cell-surface antigen 2 (Trop-2); (3) LIV-1; (4) the mucin 1-attached sialoglycotope CA6.

GPNMB was found highly overexpressed in aggressive tumors like TNBC, or in advanced setting, where it is involved in processes like cell migration, invasion, angiogenesis, or epithelial-mesenchymal transition [176,177,178]; in addition, it represents a biomarker of poor prognosis in breast cancer [179]. It is the target of Glembatumumab vedotin (CDX-011) a potent ADC conjugated with the microtubule-disrupting agent monomethyl auristatin E (MMAE) [180]. The phase II EMERGE trial, designed to evaluate CDX-011 activity in advanced GPNMB-expressing breast cancer versus chemotherapy of the investigator’s choice, showed that this drug is well tolerated and more effective in patients with TNBC and/or GPNMB-overexpressing breast cancers [181]. The following pivotal phase II trial METRIC, designed to evaluating CDX-011 versus capecitabine in TNBC GPNMB-over-expressing patients, confirmed results of safety from EMERGE trial, but the primary end point of PFS was not met [182,183]. Nonetheless GPNMB remain a potentially useful target for other ADCs agents.

Trop-2 is a type I transmembrane glycoprotein, with a relevant role in migration, cell proliferation, cell cycle progression, and metastasis [184,185,186,187]. Sacituzumab govitecan (IMMU-132) is the new promising antibody targeting Trop-2, linked to topoisomerase-I inhibitor SN-38, the active metabolite of irinotecan that induces DNA damage [188,189]. Results from a phase I/II IMMU-132-01 (NCT01631552) showed the efficacy of Sacituzumab Govitecan-hziy, with 33.3% ORR in heavily pretreated TNBC patients [190]. Based on these results, on 22 April 2020, the FDA granted accelerated approval to Sacituzumab Govitecan-hziy for adult patients with metastatic TNBC who received at least two prior therapies for metastatic disease [190]. Currently, the randomized Phase III ASCENT clinical trial, comparing Sacituzumab Govitecan-hziy versus treatment of physician’s choice, in metastatic TNBC patients who progressed after at least two prior cytotoxic therapies, is ongoing (NCT0257445599). 

LIV-1 is a zinc transporter protein downstream target of STAT3, implicated in cell adhesion and epithelial-to-mesenchymal transition [191,192,193,194]. Its target therapy, the monoclonal antibody against the extracellular domain of LIV-1, Ladiratuzumab vedotin (SGN-LIV1A), showed high efficacy in preclinical models [195], and is under evaluation in patients with metastatic breast cancers, with promising results in metastatic TNBC (NCT03310957, NCT01969643, NCT04032704, NCT03424005, NCT01042379) [196].

CA6 is selectively expressed on solid tumors and is, therefore, an ideal target for ADC therapy. SAR566658 is an ADC directed against CA6 which carries DM4, a maytansine-derived anti-microtubule agent as payload. Based on promising results from a phase I trial, a phase II study in CA6-positive TNBC (NCT02984683) is currently ongoing [197].

### 5.2. Circulating Tumor Cells as Prognostic and Predictive Biomarkers in TNBC

Detection of circulating tumor cells (CTCs) has a promising potential as minimally invasive “liquid biopsies” that can facilitate prognosis, target therapy, or monitoring therapeutic response to drugs in several cancers [198,199]. CTC counts in cancer patients have been used as a dynamic prognostic biomarker in both early and metastatic cancer [200,201], while isolation and analysis of CTCs have been shown to provide information on dynamic changes in tumor [202,203].

Over the past decade, several strategies were developed to capture CTCs based on biological properties of the cells, like the expression of cell surface proteins, or biophysical properties using filtration, microfluidics, and dielectrophoresis, or by applying high throughput imaging to unpurified blood cell preparations [204]. Despite this, there are still certain technological limitations related to sensitivity and specificity, and a lack of consensus regarding the isolation technique to be used, the type of sample, the conditions of collection or storage, or the candidate biomarker to be used [205]. At present, the only one approved for application in the clinical practice is the CellSearch technology (Menarini Silicon Biosystems, Huntingdon Valley, PA, USA), based on epithelial cell adhesion molecule (EpCAM)-based CTC isolation technology, which was FDA patented on 2004 [206,207].

A consistent number of prospective studies have demonstrated that CTC counts in cancer patients can be used as a dynamic prognostic biomarker in metastatic disease. More than a decade ago, Cristofanilli et al. in a study involving 177 metastatic breast cancer (MBC) patients, demonstrated that CTC count detected using the CellSearch was an independent prognostic factor for PFS and OS in metastatic Breast Cancer. The cut-off of 5 CTCs/7.5 mL was identified to classify patients with good or poor clinical outcome [208] and subsequent studies have confirmed the prognostic value of CTCs with the same cut-off [202,209,210,211,212]. Moreover, other studies indicated that CTCs dynamics seem to reflect treatment response as an indicator to monitor the effectiveness of treatments and guide subsequent therapies in breast cancer [213]. Other studies revealed that CTCs counts loose its prognostic value in MBC treated with targeted therapies in HER2 positive tumors [214,215]. A recent large, retrospective study, involving 1944 MBC patients showed as CTCs enumeration should be used for prognostic stratification of MBC in two defined group of patients identified as Stage IV indolent and Stage IV aggressive, where Stage IV aggressive could better benefit from novel therapies compared with Stage IV indolent [216]. In early breast cancer, CTCs count is also a prognostic biomarker, not correlated with the other usual prognostic factors. The presence and also the quantity of CTCs has proven to be associated with worse outcome, however, CTC detection methods with higher sensitivity could be necessary considering the low number of cells found in early setting [217,218,219,220,221,222,223]. 

Several groups reported that the use of CellSearch platform for CTCs counts has limited prognostic power in TNBC, because of the fact that in these tumors, cells tend to switch from epithelial to mesenchymal phenotype, loosing EpCAM expression and exhibiting more stem cell-like properties [224,225,226]. Moreover, data from clinical trials, evaluating the prognostic and predictive role of CTCs in TNBC, are controversial. Munzone et al. retrospectively analyzed the CTC enumeration by CellSearch in 203 patients with MBC, and found that baseline CTC counts were significantly associated with OS but not with PFS in TNBC patients who were receiving new courses of systemic therapy [227]. CBCSG004 trial showed that baseline CTCs count is a prognostic but not a predictive factor to anticancer therapies in TNBC [228]. The phase II trial (TBCRC019) has analyzed if CellSearch was effective in TNBC, and whether CTC apoptosis and CTC clusters enhances the prognostic role of CTC in metastatic TNBC patients treated with nab-paclitaxel with or without tigatuzumab. Results showed that patients with elevated CTC at baseline, on day 15 and 29, had significant worse PFS versus not elevated, while there was no apparent prognostic effect comparing CTC apoptosis versus non-apoptosis, or the presence of CTC clusters [229]. The large, prospective, randomized study SWOG S0500, showed that patients with TNBC and low CTC levels at baseline, and those who had CTC clearance after chemotherapy treatment, had a longer OS compared with those who had elevated CTC levels [230]. A prospective study in TNBC comparing CellSearch and immunomagnetic enrichment/flow cytometry methods revealed that CTC enumeration by two different assays was highly concordant. Both assays showed an association between baseline CTC levels and OS, and changes in CTC levels during chemotherapy were significantly associated with time to progression and OS [231]. 

CTC analysis of the tnAcity trial reported better outcomes among patients with CTC levels at baseline that were reduced or eliminated in subsequent cycles of chemotherapy, compared with CTC levels that persisted post-baseline, suggesting that CTC clearance may predict the chemosensitivity of metastatic TNBC tumors. However, as reported by the authors, this study has some limitations because of the use of only CellSerch platform that can exclude populations of low- or non-EpCAM-expressing cells, the possibility that during chemotherapy cells underwent an epithelial to mesenchymal transition change, and the low number of patient samples (N = 126) [232]. Finally, Liu et al. developed a combined CTC-NK enumeration strategy that allows to predict PFS in TNBC. They reported that baseline CTC counts can predict PFS only in first-line TNBC patients but not in other TNBC lines of therapy, while baseline CTC combined with NK enumeration (CTC-NK) can predict PFS of TNBC patients regardless of their lines of therapy [233]. The 2019 AACR annual meeting reported a significant correlation between high levels of co-expression of CCR5 and HER2 in CTCs in MBC. CCR5 has been associated with cancer stem cells and believed to drive metastatic process. The researchers suggest that identifying CCR5 expression in CTCs could be used as a potential new biomarker for MBC with potential therapeutic implications in patients with TNBC [234].

Single CTCs have been extensively studied in recent years because its detection could be particularly useful for certain types of cancers, however, also actively growing, aggressive tumors tend to release relatively low numbers of detectable CTCs into the circulation, and technical improvements in their isolation are needed [235]. Factors leading to the generation of CTCs from a primary tumor are unknown. The number of CTCs released in the bloodstream is enormously higher compared to the number of metastatic lesions in patients, indicating that the majority of CTCs die in the bloodstream, with only a minor fraction representing viable metastatic precursors. Mesenchymal transformation, stromal-derived factors, or persistent interepithelial cell junctions may provide survival signals that attenuate this apoptotic outcome [236,237]. Gene characterization of CTCs from TNBC revealed their attitude to epithelial—mesenchymal transition (EMT) associated with increased plasticity and aggressiveness, increase of resistance to cell death and chemotherapy, capability to metastasize and senescence [238,239,240,241,242]. Since a consistent group of TNBC patients are negative to CellSearch system, Abeu et al. used a method based on CellSearch system for enumeration and a combination of immunoisolation and gene expression profiling, to molecularly characterize the population. Gene profiling revealed the expression of hybrid EMT and stem cell markers associated with poor prognosis and high aggressiveness, such as *VIM*, *SNAIL1*, *TIMP1*, *CRIPTO1*, *CD49F*, *ALDH2*, *CD44*, and *BCL11A* [242]. Razmara et al. by using a PDOX models of triple-negative breast cancer (TNBC), showed that CTC clusters and CTCs expressing a mesenchymal marker (vimentin) were associated with metastatic burden in lung and liver [243]. While, Thangavel et al. reported that evaluation of EMT-specific signature did not show significant differences between CTC cluster+ and CTC cluster tumors, in a TNBC PDX model [244]. In particular, CTCs cluster, ranging from 2–50 cells, were detected in the circulation of patients with metastatic cancers [245,246]. Although rare compared with single CTC, CTC clusters are more efficient than individual CTCs in seeding metastatic colonies, they are more resistant to apoptosis and may have an advantage in physically lodging in the Lumia of vessels [235,238]. Several studies have detected CTC clusters in breast cancer [247,248,249] where the expression of mesenchymal markers was found, rather than in single migratory cells [250]. Moreover, most of the evidences of the clinical impact of CTC-clusters in breast cancer have been gathered from prospectively designed clinical studies on rather homogeneous and well-selected cohorts of metastatic patients [229,248,251,252,253]. 

In summary, CTCs, CTC clusters detection and their relative molecular characterization represent a significant source of biomarkers. However, in TNBC, detection methods need to be improved in order to not lose a consistent number of cells with mesenchymal phenotypes. Moreover, large trials are needed to confirm their prognostic and predictive role.

Cancer cells can also disseminate their contents as free DNA fragments and exosomes into the bloodstream via different mechanisms [254,255]. The quantitative analysis of DNA fragments in the blood called circulating tumor DNA (ctDNA) has emerged as potential prognostic biomarker to reveal the presence or absence of tumors or to predict relapse and metastasis. The identification of tumor-specific genetic alterations could be used to lead personalized therapies, while, in metastatic setting, liquid biopsy appears to be a good alternative to tissue biopsy [256]. Despite the potential benefits of the use of DNA sequencing assays, ctDNA is still far to be integrated in the clinical practice. The prognostic value of ctDNA is still under evaluation in TNBC tumors. In a retrospective cohort of 164 patients with metastatic TNBC, the presence of a cell-free DNA fraction greater than 10% was associated with worse outcomes, regardless of clinicopathological data [257]. A study from Parsons et al. of NGS analysis of plasma-derived and tissue biopsies DNA from 26 patients with metastatic TNBC, revealed a concordance of 70%, demonstrating the potential of liquid biopsies for mutational profiling and serial monitoring [258]. High rate of concordance, 75% and 100% for *PIK3CA* and *AKT1* respectively, was also found in the analysis of comparison between plasma-based and tissue-based DNA sequencing of LOTUS trial [259]. While, Vidula et al. revealed the presence of *BRCA* somatic mutations in ctDNA not detected in primary tumors materials [260]. In early setting, a prospective study on a cohort of 101 patients, showed that ctDNA levels at diagnosis was higher in TNBC compared to HER2+ and ER+/HER2− [261]. In a study on 46 TNBC patients, ctDNA levels detected by digital PCR, showed that no patient had detectable ctDNA after surgery, expect one patient who experienced tumor progression during neoadjuvant chemotherapy. Despite this, pCR rate was not correlated with ctDNA detection at any time point, while ctDNA positivity after one cycle of chemotherapy was correlated with shorter DFS and OS [262]. Furthermore, Cavallone et al. reported that ctDNA detection after neoadjuvant chemotherapy and before surgery was associated to DFS and OS [263]. A phase II clinical trial (NCT03145961) is recruiting patients to evaluate whether ctDNA detection can be used to detect residual disease after standard primary treatment in early-stage TNBC.

### 5.3. CSCs and Drug Resistance in TNBC

In the past years, numerous evidences highlighted the contribution of cancer stem cells (CSCs) in tumorigenic potential, high risk of metastasis, and drugs resistance of TNBC [264]. CSCs represent a small population of cancer cells with staminal phenotype; they expose CD44+/CD24− and high ALDH expression [265,266,267]. This signature is associated with high capability of self-renewal, of proliferation and mammalian spheroids forming [268,269,270]. It is unclear if CSCs arise from pathogenic mutations in resident stem cells, or if they are the result of mutations of quiescent cells [271], anyway these CD44+/CD24− cells show an EMT phenotype with a great tumorigenic ability invasion and metastasis [265,272,273]. In this context, TNBC seems to have a significant number of stem cells CD44+/CD24− [266], and high expression of ALDH1 [3]. Breast cancer tissues of TNBC patients have shown to express high stem cell markers [272,274,275] and the gene signature of TNBC cells seems remarkably similar to that of mammary stem cells TNBC [264]. In an analysis of 466 invasive breast carcinomas and eight breast cancer cell lines, basal-like breast cancer harbored the highest percentage of tumor cells with the CSC phenotype CD44+CD24−/low and ALDH1 positivity [276]. In clinical studies, CD44+CD24−/low expression was associated with worse chemotherapy response, lymph node metastasis, distant metastasis, recurrence, and worse DFS and OS [277,278]; while ALDH1-expression predicted poor prognosis in TNBC patients [279,280,281]. As reported above, EMT phenotype confers capacity to metastasize and drug resistance in TNBC, and several researches have highlighted that EMT transition and enrichment for CSCs in TNBC tumors were correlated to invasiveness and drug resistance [282,283,284].

In particular, it seems that over-expression of EMT pathways promote the generation of mammary CSCs and is responsible of capacity of CSCs to survive in hard metabolic conditions because of reduction of nutrients or hypoxia [282,285,286]. 

Several evidences have reported that self-renewal activity of CSCs could be ascribed to the alteration of different signal transduction pathways such as STAT signaling, SRC signaling, Wnt/β-catenin signaling. In particular, STAT3 signaling is involved in the mechanism for self-renewal regulation of CSCs, and conversion of non-CSCs into CSCs, through the regulation of IL-6-Jak1-STAT3-OCT3 [287] or in tumorigenic potential, mammosphere-forming efficiency, and ALDH activity of breast cancer cells through VEGFR-2/STAT3 signaling [288]. 

In TNBC patients, STAT3 activation is a biomarker of poor prognosis. The phosphorylated isoform is preferentially expressed in TNBC cell lines [289], and it seems associated to initiation, progression, metastasis, and chemotherapy resistance [290]. In this regard, STAT3 signaling inhibitors are in clinical trials evaluation also in TNBC patients [291,292,293]. 

## 6. Discussion

TNBC is an intrinsically heterogeneous group of breast cancer and there is a need for effective biomarkers that can help physicians in selecting the most appropriate treatment.

Several proposed biomarkers for TNBC have been studied in clinical trials demonstrating, so far, modest clinical benefits. Mutations of *BRCA1/2* genes turned out to be factors predicting the efficacy of PARPis and alterations of other genes involved in homologous recombination seem promising in this setting. 

PD-L1 protein expression either in IC, tumor cells or both can be used as a predictive biomarker for response to immunocheckpoint inhibitors. There are different commercially available assays that use different antibodies as long as different scoring systems. However, there still is uncertainty as to what is the best assay in TNBC and if the results apply to all immunocheckpoint inhibitors. At present, however, data clearly support the use of the VENTANA SP142 assay to predict the efficacy of Atezolizumab in metastatic TNBC.

Table 1 summarizes the main discussed biomarkers and their prognostic and predictive significance.

## 7. Conclusions

Despite many research efforts, only a few useful biomarkers have been identified so far in TNBC. Some of these are already in the clinical practice. As new therapeutic agents are developed, parallel preclinical and clinical research is needed to identify biomarkers for the responsive, or conversely, the resistant patient population.

## Figures and Tables

**Table 1 ijms-21-04579-t001:** Summary of biomarkers in triple negative breast cancer.

Biomarker	Main Function	Assay	Prognostic/Predictive Significance	Target Therapy	Ref.
*BRCA1* and *BRCA2* genes	DNA-double strand break repair	BRACAnalysis CDx test HRDetect assay HRD assay myChoice CDx	Poor prognostic factor. High response to platinum-based therapy and predictor for response to PARP inhibitors	PARP inhibitors	[9,10,30,55,58,59,60,63,64,65]
HRR genes	Homologous recombination repair of DNA		Predictor of response to platinum therapy in neoadjuvant setting	ATR inhibitor * WEE1 inhibitor *	[41,47,56,57,59,60,61,62]
Stromal TILs	Tumor infiltrating lymphocytes involved in immune response against the tumor	Tissue Immunohistochemistry	High TILs correlates with more favorable survival outcomes and are predictive for increased response to neoadjuvant CT	NA	[92,93,94]
PD-L1 protein	Tumor immune evasion process	VENTANA PD-L1 (SP142) Assay	High expression correlates with higher survival rates in trials with ICI	Immune checkpoint inhibitors	[11,117,118,119,120,121,122,123,124,125]
Microsatellite instability (MSI)	High Immunogenic activity	Histologically/cytologically confirmed MSI-H/dMMR	Predictor of response to Pembrolizumab	Pembrolizumab	[126]
PI3-kinase pathway	Cell proliferation	Tissue Immunohistochemistry of PI3KCA/PTEN or PI3k pathway genomic sequencing by NGS	Higher sensitivity to AKT inhibitors and to combination therapy of PI3K and androgen receptor inhibitors in LAR tumors	PI3K inhibitor * AKT inhibitor *	[163,168,169,170,171,173]
GPNMB	Cell migration, invasion, angiogenesis, epithelial-mesenchymal transition	Tissue Immunohistochemistry	Poor prognostic factor	Glembatumumab vedotin (Antibody-drug conjugate) *	[179,181,182,183]
Trop-2	Cell cycle progression, migration, proliferation, metastasis	Tissue Immunohistochemistry	Poor prognostic factor	Sacituzumab Govitecan-hziy (Antibody-drug conjugate) *	[189,190]
LIV-1	Cell adhesion, epithelial-mesenchymal transition	Tissue Immunohistochemistry	Under investigation	Ladiratuzumab vedotin (Antibody-drug conjugate) *	[196]
CA6	Tumor cell survival and proliferation	Tissue Immunohistochemistry	Under investigation	SAR566658 (Antibody-drug conjugate) *	[197]

* Under clinical investigation; ICI: immune checkpoint inhibitors.

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
