# Peer review of "Biomarkers in Triple-Negative Breast Cancer: State-of-the-Art and Future Perspectives"

_ijms, 2020, doi:10.3390/ijms21134579_

Round 1

Reviewer 1 Report

The authors provide a comprehensive review of TNBC biology and potential therapies. There are several suggestions that could make the work more useful for a general audience.

  1. The introductory paragraph acknowledges the approval of atezolizumab in the metastatic setting (line 47). Pembrolizumab in the adjuvant setting (Keynote 522) should also be mentioned.
  2. In the discussion of PARPi, some statements regarding whether somatic BRCA mutations can identify sensitive tumors should be mentioned.
  3. The HRD assay (Myriad) assay is mentioned as a potential predictive biomarker for PARPi sensitivity, but some discussion of sensitivity to platinum agents as identified by this assay should be discussed.
  4. Overall, there is little discussion about platinum agents in TNBC. Since this is an important clinical topic with numerous completed trials in the neoadjuvant, adjuvant, and metastatic setting reported, some discussion of this class of chemotherapy should be included in an otherwise comprehensive TNBC review.
  5. When the authors discuss the controversy about PDL1 testing (line 237) some mention of which cells are predictive (tumor versus IC) and assay differences should also be mentioned. There is little discussion of tumor mutational burden, microsatellite instability, or DNA mismatch repair either.
  6. The discussion of LAR TNBCs should be addressed under a separate heading from the PI3K section. While there may be synergy between AR and PI3K targeting, the inclusion of this section in the middle of the discussion of PI3K targeting seems to interrupt the flow of the argument.
  7. In the section of the PI3K inhibitors, there is a discussion of the FAIRLANE trial (line 335), but a parallel discussion of PARPi (or platinum) in the neoadjuvant (Litton, et al. J Clin Oncol 38:388 2020 PMID: 31461380) setting seems to be missing.
  8. The FDA approval of Sacituzumab govitecan needs to be mentioned with update of the data that led to the approval (Bardia, et al. N Engl J Med 380:741 2019 PMID: 30786188).
  9. A comprehensive discussion of CTCs is included, but no mention of ctDNA. If the authors are interested in explaining how serum biomarkers may be useful in TNBC, some mention of ctDNA needs to be included.
  10. The manuscript itself is fairly dense. A table outlining the separate molecular subtypes, targets, and referral to appropriate clinical data would be useful.

Author Response

The authors provide a comprehensive review of TNBC biology and potential therapies. There are several suggestions that could make the work more useful for a general audience.

  1. The introductory paragraph acknowledges the approval of atezolizumab in the metastatic setting (line 47). Pembrolizumab in the adjuvant setting (Keynote 522) should also be mentioned.

From our knowledge, Pembrolizumab has not yet been approved in early breast cancer. In the introduction we mentioned approved ones only.

  1. In the discussion of PARPi, some statements regarding whether somatic BRCA mutations can identify sensitive tumors should be mentioned.

A statement has been added as per request on lines 80-82, 162-175.

  1. The HRD assay (Myriad) assay is mentioned as a potential predictive biomarker for PARPi sensitivity, but some discussion of sensitivity to platinum agents as identified by this assay should be discussed.

The discussion of predictive value of HRD assays was reported in the section of platinum chemotherapy in BRCA chapter, on 184-228 lines.

  1. Overall, there is little discussion about platinum agents in TNBC. Since this is an important clinical topic with numerous completed trials in the neoadjuvant, adjuvant, and metastatic setting reported, some discussion of this class of chemotherapy should be included in an otherwise comprehensive TNBC review.

A section on platinum chemotherapy was included in BRCA chapter (176-245 lines).

  1. When the authors discuss the controversy about PDL1 testing (line 237) some mention of which cells are predictive (tumor versus IC) and assay differences should also be mentioned. There is little discussion of tumor mutational burden, microsatellite instability, or DNA mismatch repair either.

Regarding the predictive value of IC versus TC a sentence was added on lines 309-311, while we suppose to exhaustively reported the assays on lines 323-347.

A discussion on tumor mutational burden, microsatellite instability, or DNA mismatch repair was added on lines 348-399.

  1. The discussion of LAR TNBCs should be addressed under a separate heading from the PI3K section. While there may be synergy between AR and PI3K targeting, the inclusion of this section in the middle of the discussion of PI3K targeting seems to interrupt the flow of the argument.

The short discussion on LAR tomours was inserted in the Introduction, lines 43-45.

  1. In the section of the PI3K inhibitors, there is a discussion of the FAIRLANE trial (line 335), but a parallel discussion of PARPi (or platinum) in the neoadjuvant (Litton, et al. J Clin Oncol 38:388 2020 PMID: 31461380) setting seems to be missing.

We have included neoadjuvant PARPi study on lines 108-118 and neoadjuvant platinum studies in the relative section.

  1. The FDA approval of Sacituzumab govitecan needs to be mentioned with update of the data that led to the approval (Bardia, et al. N Engl J Med 380:741 2019 PMID: 30786188).

A mention has been added on line 546.

  1. A comprehensive discussion of CTCs is included, but no mention of ctDNA. If the authors are interested in explaining how serum biomarkers may be useful in TNBC, some mention of ctDNA needs to be included.

ctDNA section was included on 668-694 lines.

  1. The manuscript itself is fairly dense. A table outlining the separate molecular subtypes, targets, and referral to appropriate clinical data would be useful.

A table (Table 1) was included with main discussed biomarkers, the assays used for their detection, their prognostic and predictive value and relative references

Reviewer 2 Report

The authors presented the review on the title “Biomarkers in Triple negative breast cancer: state of the art and future perspectives”.

Breast cancer is a common cancer and is the leading cause of cancer-related deaths among women worldwide and estrogen receptor (ER) positive breast cancer accounts for 70% breast cancer subtype.

Triple negative breast cancer (TNBC) is a subtype of breast cancer lacking of expression of estrogen receptor (ER), progesteron receptor (PgR) and Human Epidermal Growth Factor Receptor 2 (HER2).

Taking these facts into consideration, there is a need to summarize the current biomarker approach discuss future exploration. In fact, the term “BRCAness” has been used to describe a dysfunctions in the BRCA-related DNA reapair mechanism that is not due to mutations of the BRCA1/2 genes.

Line 115, there is typo in the name of the Researcher, the correct one is: Domagala.

TNBC is characterized by high mutation rate and greater tumor-infiltrating lymphocytes (TILs). That is very important, good the Authors pay attention on this point.

Authors may wish add some additional references to confirm the thesis (lines: 177-180) that effect of chemotherapy might be partially mediated by the immune system. In my opinion this merits for broader research.

The Authors made extensive work, a conclusion fully support investigation made.

In terms of language, the article English is fine.

Author Response

The authors presented the review on the title “Biomarkers in Triple negative breast cancer: state of the art and future perspectives”.

Breast cancer is a common cancer and is the leading cause of cancer-related deaths among women worldwide and estrogen receptor (ER) positive breast cancer accounts for 70% breast cancer subtype.

Triple negative breast cancer (TNBC) is a subtype of breast cancer lacking of expression of estrogen receptor (ER), progesteron receptor (PgR) and Human Epidermal Growth Factor Receptor 2 (HER2).

Taking these facts into consideration, there is a need to summarize the current biomarker approach discuss future exploration. In fact, the term “BRCAness” has been used to describe a dysfunctions in the BRCA-related DNA reapair mechanism that is not due to mutations of the BRCA1/2 genes.

Line 115, there is typo in the name of the Researcher, the correct one is: Domagala.

Corrected, refer to line 134.

TNBC is characterized by high mutation rate and greater tumor-infiltrating lymphocytes (TILs). That is very important, good the Authors pay attention on this point.

Authors may wish add some additional references to confirm the thesis (lines: 177-180) that effect of chemotherapy might be partially mediated by the immune system. In my opinion this merits for broader research.

Two additional references were included (92, 93). Refer to line 284.

The Authors made extensive work, a conclusion fully support investigation made.

In terms of language, the article English is fine.

Round 2

Reviewer 1 Report

The authors have been very responsive to previous comments. The manuscript will be of interest to a broad audience as a comprehensive review of the topic.